

# Opinion: The importance and future development of perturbed parameter ensembles in climate and atmospheric science

Ken S. Carslaw[1], Leighton Regayre[2], Ulrike Proske[3], Andrew Gettelman[4], David M. H. Sexton[5], Yun Qian[4], Lauren R. Marshall[6], Oliver Wild[7], Marcus van Lier-Walqui[8], Annika Oertel[9], Saloua Peatier[10], Ben Yang[11], Jill S Johnson[12], Sihan Li[13], Daniel T. McCoy[14], Benjamin M. Sanderson[15], Christina J. Williamson[16], Gregory S. Elsaesser[17], Kuniko Yamazaki[5], Ben B. B. Booth[5].

1. School of Earth and Environment, University of Leeds, UK.
2. School of Earth and Environment, University of Leeds, UK and Met Office, Exeter, UK.
3. Hydrology and Environmental Hydraulics Group, Wageningen University, Wageningen, The Netherlands.
4. Pacific Northwest National Laboratory, Richland, WA, USA.
5. Met Office, Exeter, UK.
6. School of Earth and Environmental Sciences, University of St Andrews, St Andrews, UK.
7. Lancaster Environment Centre, Lancaster University, Lancaster, UK.
8. NASA Goddard Institute for Space Studies, New York, NY, USA and Center for Climate Systems Research (CCSR), The Earth Institute, Columbia University, NY, New York, USA.
9. Karlsruhe Institute of Technology, Institute of Meteorology and Climate Research Troposphere Research, Karlsruhe, Germany.
10. Department of Meteorology, University of Reading, UK.
11. School of Atmospheric Sciences, Nanjing University, Nanjing, China.
12. School of Mathematical and Physical Sciences, University of Sheffield, Sheffield, UK.
13. School of Geography and Planning, University of Sheffield, Sheffield, UK.
14. Department of Atmospheric Science, University of Wyoming, Laramie, WY, USA.
15. CICERO Center for International Climate Research, Oslo, Norway.
16. Finnish Meteorological Institute, Helsinki, Finland and Institute for Atmospheric and Earth System Research/Physics, University of Helsinki, Finland.
17. Department of Applied Physics and Mathematics, Columbia University, New York, NY, USA and NASA Goddard Institute for Space Studies, New York, NY, USA.

**Correspondence to**: Ken S. Carslaw (k.s.carslaw@leeds.ac.uk)

**Abstract.** A grand challenge in climate science is to translate advances in our fundamental understanding into reduced uncertainty in climate projections Model uncertainty, characterized for example by the spread of simulations of future climate projections, has changed little over the past few decades despite major advances in model complexity, resolution, and the growing number of intercomparison projects and observational datasets. Here we argue that the use of perturbed parameter ensembles (PPEs) would accelerate our understanding of uncertainty in its broadest sense and help identify strategies for reducing it. We make eleven recommendations for future research priorities, drawing on existing studies that use PPEs to guide model development and simplification, understand inter-model differences, more fully characterize the plausible spread in



climate projections, formalize model calibration, define observational requirements, and investigate how interacting environmental conditions influence complex climate systems like cloud fields. These studies extend across climate, weather,

atmospheric chemistry, clouds, aerosols and renewable energy using process-based high-resolution models through to global-scale models. Although increases in model complexity, resolution and intercomparison projects consume most computing resources today, we argue that, in synergy with these efforts, PPEs are essential for fully characterizing model uncertainty and improving model reliability, and that they should be prioritized when allocating those resources.

## 1. The challenges of climate modelling

The future evolution of atmospheric, climate and Earth system models involves several well-motivated yet partly competing priorities, each placing increasing demands on computing resources. Most model development is motivated by the goal of improving model fidelity – the extent to which model simulations reproduce the observed state and behaviour of the climate and Earth system. To that end, the community has focused primarily on increasing process-level detail (complexity) and spatial resolution, as well as on running initial-condition ensembles to characterize decadal and regional climate variability by

sampling internal variability. This effort has produced impressive model simulations that look and behave in many ways like the real world.

The other vital aspect of modelling alongside fidelity is reliability – the extent to which models produce consistent and trustworthy results across multiple scenarios. The primary approach used to assess model reliability is the model intercomparison project (MIP), where the spread of simulations serves as a rough, yet incomplete, proxy for reliability (see

Table 1 for a glossary of uncertainty terms used in this article). The number of climate MIPs has grown from two standardized experimental protocols in the 1990s to 322 in phase 6 of the Coupled Model Intercomparison Project (CMIP) in 2017 (Durack et al., 2025). MIPs have revealed systematic model biases, improved our understanding of the climate system, and guided international assessments and policy. However, MIPs represent an incomplete and unquantified mixture of model structural differences, parametric uncertainties and internal variability (Knutti et al., 2010).

The incompleteness of MIPs has two important consequences. Firstly, MIPs represent the main way that we communicate plausible uncertainties to wider science and impact communities – for example through a subset of CMIP simulations that inform IPCC's Impacts, Adaptation and Vulnerability assessments (WG2). An incomplete representation of uncertainties may therefore have conveyed an unjustifiably high level of confidence, inconsistent with the underlying physical modelling. The second consequence relates to how we move forward. Without a full understanding of the magnitude and causes of model

spread, we lack the information required to improve model reliability, assuming this is possible.

Although the community emphasizes model complexity, greater complexity does not equate to greater reliability (Baartman et al., 2020; Knüsel and Baumberger, 2020; Proske et al., 2024; Puy et al., 2022). This is essentially the over-fitting problem – see Fig. 1. Over more than 30 years and six phases of CMIP, increasingly detailed components of the climate system have been evaluated and compared across models (Durack et al., 2025). Yet, for key emergent climate metrics—such as climate

sensitivity, cloud feedback, and aerosol radiative forcing—the spread among models has remained substantial. The famous





quote from George Box "All models are wrong, but some are useful" was followed by perhaps the more relevant statement that "The scientist cannot obtain a 'correct' [model] by excessive elaboration" (Box, 1976; Carslaw et al., 2018). Climate model spread might persist or even grow as we obtain new knowledge (Knutti and Sedláček, 2013), and we may at present even be underestimating the spread. But it is also the case that efforts to enhance model complexity have not been matched by

efforts to understand and reduce the uncertainty that it introduces. In short, our understanding of model uncertainty is incomplete, and we are certainly not on a path to reducing it.

Increases in model resolution (Slingo et al., 2022) will further enhance model fidelity. However, some critical processes will always be parameterized even in km-scale models (Morrison et al., 2020), which will continue to affect model reliability. Furthermore, any high-resolution MIP to assess reliability would still represent an unquantified mixture of structural

differences, parametric uncertainties and internal variability, making assessment of reliability perhaps even more challenging.

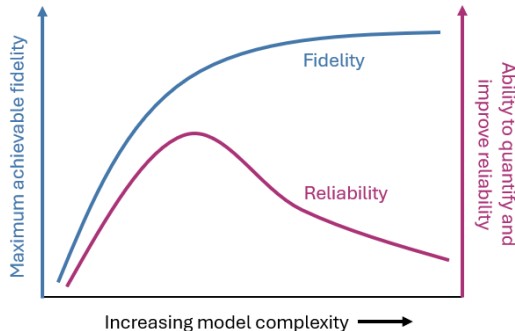

**Figure 1. Schematic of the model complexity-reliability challenge. Models with greater complexity can in principle achieve higher fidelity. However, higher-complexity models have a larger number of uncertain processes in them, which become increasingly difficult to verify against observations. This is often referred to as over-fitting. Complex models**
**are also slower to run, which limits how the tools of uncertainty quantification and reliability assessment, such as PPEs, can be applied.**

What is the role of perturbed parameter ensembles? The prevailing view is that the uncertainty caused by adjustable quantities inside a model (the parameters – see Table 1) is just a small part of the overall uncertainty challenge – mainly just a component

of model tuning. While tuning is essential and universal (Hourdin et al., 2017), expansion into full perturbed parameter ensembles (PPEs) in which combinations of parameters are systematically perturbed, is given much lower priority than structural changes to models. When a model is structurally deficient—due to low resolution or missing processes—addressing these issues is prioritized, while PPEs are seen as merely "over-polishing" a structurally deficient model. This narrow perspective is one we challenge here.

In this article, we highlight how PPE science has expanded since the 2015 Workshop on Uncertainty Quantification in Climate Modeling and Projection (Qian et al., 2016) to address the *overall* challenge of model uncertainty and to provide an increasingly deep, process-level understanding of model behaviour. In particular, we argue that PPEs can help to more fully characterize





current model uncertainty and provide a path toward reducing it. We start with a brief introduction to PPEs before summarizing the range of recent PPE studies in atmospheric and climate science, how they are contributing to wider climate science challenges, and how these efforts could be further developed.

**Table 1. Glossary of uncertainty terms referred to in this article. Terms in italic are defined elsewhere in the table.**

| Term | Definition |
|---|---|
| Internal variability | Uncertainty in model simulations arising from natural fluctuations in the climate system, even when external forcings, the model structure and *parameters* are fixed. Its size can be estimated by initial condition ensembles (Maher et al., 2021). |
| Parameter | Adjustable quantity inside a model, most often within the model parameterization equations. |
| Parametric uncertainty | Uncertainty in model output variables caused by uncertainty in the values of *parameters* of the model, when the structure of the model (equations and processes) is fixed.<br>In *perturbed parameter ensembles*, parameters have been expanded to include perturbation of environmental boundary conditions and other inputs like topography. |
| Structural uncertainty | Uncertainty arising from the way a model is built – its structure, equations, resolution, assumptions, and simplifications rather than the specific values of *parameters* or inputs. |
| Epistemic uncertainty | Uncertainty in a model's predictions stemming from what we don't know about the system being modelled, including uncertainty in structural choices, process representations and *parameter* values. |
| Perturbed parameter ensemble (PPE) | A set of model simulations created by systematically varying uncertain *parameters* within a model to explore the effect on model behaviour. PPEs are 'designed' ensembles in that the variations in output are due to defined differences between the ensemble members. |
| Multi-model ensemble (MME) | Simulations of a set of models usually following a prescribed 'experimental protocol', although not 'designed' in the same way as a PPE – see *perturbed parameter ensemble*. |
| Fidelity | The extent to which model simulations reproduce the observed state and behaviour of the climate system, in particular how well the model captures these in sufficient detail. |
| Model spread (sometimes called diversity) | The range of outcomes produced by a set of climate models in response to the same scenario or inputs.<br>Model spread in a MME does not fully represent the model *reliability* because it is an unquantified mixture of model *structural uncertainty*, *parametric uncertainty* and *internal variability* unless these are standardized or suppressed in the experimental design. |
| Reliability | The extent to which models produce consistent and trustworthy results across multiple scenarios or applications contexts.<br>A proxy for reliability is the *model spread* in a *multi-model ensemble*, but it is incomplete because the models do not represent the true range of *epistemic uncertainty*. Reliability can manifest differently when observations are available to |





|  |  |
|---|---|
|  | test *model performance*, as opposed to when making projections, where models may diverge despite good *model performance* due to model *equifinality*. |
| Robustness | Stability of model results across varying assumptions or uncertainties in the model. |
| Model performance | The goodness of fit or agreement of a model with observations. |
| Structural deficiency | A systematic limitation or flaw in the way a model represents the underlying processes (e.g., incomplete, wrong or over-simplified) leading to reduced realism. |
| Equifinality | The concept that different models, or different combinations of model parameters and structures within a model, can produce equally good *model performance*. Different models may diverge when making predictions, introducing previously hidden uncertainty and unreliability. |
| Constrain | To reduce uncertainty in model simulations. Constraint does not mean improvement of *model performance,* but a narrowing of the quantified uncertainty range, for example using methods of *history-matching*. |
| History-matching | A systematic, statistical process of finding combinations of *parameter* settings (*model variants*) that match observations within their uncertainty range. Unlike *tuning*, history matching starts with a dense sample of model *parametric uncertainty* and aims to rule out observationally implausible model variants rather than to find a single "best" model. |
| Model variant | In PPEs, a model variant is a specific version of a model with different parameter choices. More widely, the term may encompass different choices of specific model process representations within the same overall model structure or family. |
| Calibration | A systematic, statistical process of adjusting model *parameters* to find the model variant(s) that best match observations within their uncertainty range. Unlike *history-matching*, calibration aims to match observations as closely as possible to improve *model performance*. Unlike *tuning*, calibration usually starts with a sampling of model uncertainty. It may result in several model variants with similar *model performance* because of *equifinality*. |
| Tuning | In contrast to *calibration* and *history matching*, tuning is often a more informal or heuristic process of adjusting model *parameters* to improve *model performance* without first sampling model uncertainty. |

## 2.   A PPE primer

A perturbed parameter ensemble (PPE) is a set (ensemble) of model simulations in which each simulation has a different combination of selected parameters. In a traditional PPE, the parameters are most-often quantities in the model's defining equations (parameterizations) and the purpose is to determine how combinations of parameter perturbations affect the model outputs. A PPE therefore explores the joint effects of changes in several parameters across a multi-dimensional 'parameter space' that cannot be learned by adjusting one parameter at a time. The increasingly diverse application of PPEs in climate

and atmospheric science originated in pioneering work on idealised experiments that doubled $CO_2$ concentrations (Murphy et al., 2004; Stainforth et al., 2005).



The PPE approach has been extended to include a wider range of quantities that can control the model outputs, such as gas and aerosol emission factors and, for some types of model set-up, boundary conditions (or 'forcings') like sea surface temperature or humidity. One benefit of this hybrid approach is that model behaviour and uncertainty can be explored in a consistent way over a range of environmental conditions (Wellmann et al., 2018).

PPEs are often combined with statistical (Lee et al., 2011; Qian et al., 2018) or machine learning (Elsaesser et al., 2025; Gettelman et al., 2024) emulation to create a much larger sample of input/output combinations for statistical analysis (Fig. 2). This is often necessary due to the high computational cost of the model, which limits simulations to a very sparse coverage of the multi-dimensional parameter space. Emulators are trained to learn the relationship between parameter settings and model outputs and, once trained, are fast to compute. Emulators enable model outputs to be estimated, ideally with associated emulator uncertainty, for any combination of parameter settings within the parameter space, which then provides a way to scale up from, say, 100 PPE members to millions of evaluations for statistical analysis. Typically, an ensemble of around 5-10 simulations per parameter produces a useful emulator, making PPEs a highly efficient computational method for exploring model behaviour. New ML/AI methods are enabling emulators to be built efficiently, reducing the cost of sampling parameter space for PPEs (Elsaesser et al., 2025; Gettelman et al., 2024).

Although PPEs are distinct from initial-condition ensembles (Maher et al., 2021), a PPE may sample internal variability, and methods have been developed to account for it in building emulators (Rostron et al., 2020; Sansom et al., 2024).

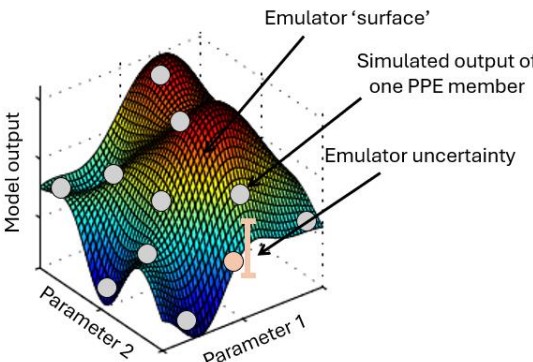

**Figure 2. A perturbed parameter ensemble and an emulator trained on the data to describe a mapping between the input parameter values and the model outputs across a multi-dimensional parameter space. The emulator can then be used to generate millions of 'model variants' for any combination of parameter values for statistical analysis.**





## 3. What science challenges are being tackled using PPEs?

This article is partly based on discussions between around 70 participants at the World Climate Research Programme workshop on the Analysis of PPEs in Atmospheric Research (APPEAR) in 2022. The APPEAR workshop demonstrated an enormous breadth of research using PPEs across over twenty models covering climate, aerosols, atmospheric chemistry, clouds and meteorology over spatial scales from large-eddy models through to regional weather and global climate models – see Fig 3.

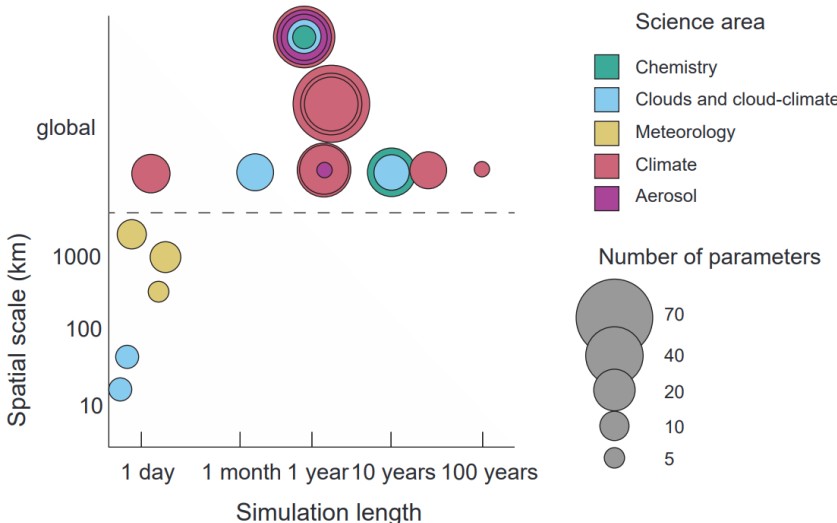

**Figure 3. PPEs in climate and atmospheric science presented at the WCRP APPEAR workshop. The area of each point scales with the number of parameters that were perturbed. The simulation length represents the time horizon of the simulations of each individual PPE member.**

In *global climate*, PPEs cover timescales of a few days to 100 years with 3 to more than 70 parameters perturbed. PPEs have been used to:

- Understand and constrain uncertainty in climate sensitivity (Brown et al., 2025; Hourdin et al., 2023; Peatier et al., 2022; Shiogama et al., 2014; Yokohata et al., 2010), energy budgets (Yang et al., 2023), precipitation (Jiang et al., 2025; Qian et al., 2015), and cloud forcing and feedbacks (Duffy et al., 2023; Eidhammer et al., 2024; Furtado et al., 2023; Gettelman et al., 2024; Tsushima et al., 2020; Zhang et al., 2018).
- Constrain climate projections and future warming rates (Peace et al., 2020; Watson-Parris, 2025; Yamazaki et al., 2021), the latter using model performance over a range of timescales from weather to climate (Sexton et al., 2021).
- Understand regional climate (Bellprat et al., 2012; Liu et al., 2022a), regional atmospheric circulation systems (Peace et al., 2022; Zhang et al., 2023), drivers of the Atlantic Meridional Overturning Circulation (Yamazaki et al., 2024), and the causes of regional temperature and precipitation biases (Li et al., 2019).





- Define 'variants' of a model that span a range of behaviours, such as the "calibrated physics ensemble" of GISS ModelE for submission to CMIP6 (Elsaesser et al., 2025), climate model variants that span a range of climate sensitivities (Hourdin et al., 2023; Peatier et al., 2022), and samples of extreme climates to support climate adaptation (Leach et al., 2022). PPEs show that the parametric uncertainty of a single model can be similar to that of multi-model ensembles (Duffy et al., 2023).


- Define and categorise structural errors within a climate model by identifying trade-offs between errors in different locations and fields and biases that are orthogonal to those resulting from parameter perturbations (Peatier et al., 2024).

In *aerosol*, global-scale PPEs have been used to:


- Understand and quantify model parameters controlling aerosol properties (Carslaw et al., 2013; Fanourgakis et al., 2019; Hamilton et al., 2014; Lee et al., 2011, 2012, 2013) and radiative forcing (Carslaw et al., 2013; Regayre et al., 2015, 2014), and to inform wider climate model tuning efforts (Sexton et al., 2021).

- Observationally constrain uncertainty in aerosol radiative forcing (Johnson et al., 2018, 2020; Lee et al., 2016; McCoy et al., 2020; Regayre et al., 2018, 2020; Watson-Parris et al., 2020) and aerosol–cloud adjustments


(Mikkelsen et al., 2025a; Song et al., 2024). In the stratosphere, a PPE explored how combinations of eruption properties affect volcanic radiative forcing (Marshall et al., 2019). As is the case for the whole climate system, parametric uncertainty in aerosol radiative forcing in one model can be comparable to the multi-model spread (Regayre et al., 2018; Yoshioka et al., 2019).

- Quantify the effect of aerosol on climate change, including climate threshold exceedances (Peace et al., 2020) and


the climatic effect of historical volcanic eruptions using ice core sulfate measurements (Marshall et al., 2021).

- Expose model structural deficiencies in the representation of aerosols and clouds (Regayre et al., 2023).

- Simulate the aerosol effects on East Asian climate and their parametric uncertainties associated with emissions and cloud microphysics (Yan et al., 2015).

In *chemistry*, PPEs have been used to:


- Quantify the sensitivity of ozone, hydroxyl radicals and methane lifetime to atmospheric conditions, uncertain processes and emissions across three global models, identifying the cause of differing model responses (Wild et al., 2020).

- Robustly calibrate model parameters through observational constraint to measured gas concentrations (Ryan and Wild, 2021). In a regional chemistry model, PPEs were used to define the important parameters of a complex


organic chemistry scheme (Reyes-Villegas et al., 2023) and on the global scale to expose potential structural deficiencies in a secondary organic aerosol scheme (Sengupta et al., 2021).

In *cloud physics,* PPEs have been applied from large-eddy scale to global scale.



- For shallow clouds, PPEs at the large-eddy scale with model domains of tens of km and resolutions of tens of metres have exposed how combinations of cloud-controlling factors (environmental conditions such as above-cloud humidity and potential temperature) affect cloud evolution and adjustments to changes in aerosol (Glassmeier et al., 2019; Sansom et al., 2024, 2025). They have provided considerable insight into how stratocumulus cloud-field properties vary over a wide range of states and to understand the timescales of cloud response to shipping emissions (Glassmeier et al., 2021).

- For deep convective clouds, the key parameters controlling liquid and ice hydrometeors and precipitation (Johnson et al., 2015) and anvil cirrus (Hawker et al., 2021) in high-resolution (60 m to 250 m) models have been identified, as well as the relative importance of model parameters and environmental conditions for forecasting deep convection and hail (Wellmann et al., 2018, 2020). On the global scale, the response of circulation and cloud responses to surface warming has been analysed to understand the results from multi-model ensembles (Schiro et al., 2019).

- On regional climate scales, Qian et al. (2024) used a set of short PPE simulations to understand how cloud forcing depends on different sets of parameters over different regions.

In *meteorology,* PPEs have been used to*:

- Understand how environmental conditions and uncertainties in cloud microphysics affect the evolution of weather-scale phenomena. Studies with 500 m grid spacing examined how 11 environmental conditions like boundary layer height and soil moisture interact to affect the characteristics of sea breezes under dry (Igel et al., 2018) and moist (Park et al., 2020) convective conditions. At 3.3 km grid spacing a PPE has been used to quantify the relative importance of uncertainties in low-level temperature and moisture compared to uncertainty in cloud microphysical processes in numerical weather predictions of warm conveyor belts (Oertel et al., 2025). This PPE was also used to quantify sensitivities of cirrus cloud properties and water vapor transport into the upper troposphere/lower stratosphere region to cloud microphysical parameters (Schwenk et al., 2025).

- Guide model development for improved tracking of wind and solar energy, as well as the design of major field studies (Berg et al., 2021; Yang et al., 2017, 2019). These studies examined the sensitivity of turbine-height wind speeds to parameters in planetary boundary layer and surface-layer schemes, quantifying both parametric and structural sensitivities. The parametric sensitivity of solar irradiance to model parameters related to cloud and aerosol processes has also been assessed in WRF-Solar, allowing these parameters to be optimized to enhance irradiance forecasting skill by up to 33% (Liu et al., 2022b, c).

## 4.    How can PPEs contribute to the future of climate modelling?

In this 'golden age of climate modeling' (Betancourt, 2022) we argue that PPEs ought to be included alongside the increases in complexity, resolution, initial-condition ensembles and model intercomparison projects. Effort in this direction would adjust



the balance towards greater consideration of model reliability (consistent and trustworthy results) and plausible model spread
alongside existing efforts to improve model fidelity (models resembling the real world). We make eleven recommendations in
six areas:

**1. Use PPEs to understand model structural differences and deficiencies, and to define priorities for model development**
**or simplification**. PPEs provide the only means to disentangle structural and parametric causes of model–observation biases,
which would provide valuable information for prioritising model developments. For example, such separation would help to
prevent over-tuning of parameters, a practice that is likely to obscure structural deficiencies and the need for targeted model
developments. It would also help to prevent the addition of process-level details to a model where a comprehensive sampling
of parameter space might remove biases – i.e., where the need for structural changes is not supported by observational evidence.
PPEs offer a rigorous framework for identifying where model developments would be most effective, reducing reliance on the
interests or parameterization-specific expertise of development teams.
PPEs have exposed structural errors when a comprehensive sampling of the model's parameter space fails to capture observed
system behaviour (Peatier et al., 2024) and when there are inconsistencies among observational constraints (Regayre et al.,
2023). Successful efforts to understand structural differences and to detect deficiencies have so far involved global climate
models (Furtado et al., 2023; Rostron et al., 2020; Sanderson, 2011; Shiogama et al., 2014; Tsushima et al., 2020; Yokohata
et al., 2010), land surface models (Hawkins et al., 2019; McNeall et al., 2016), ocean models (Williamson et al., 2015), and
regional and global chemistry and aerosol models (Regayre et al., 2023; Reyes-Villegas et al., 2023; Sengupta et al., 2021).
Alongside the detection of model deficiencies, PPEs can also help to identify opportunities for model simplification (Proske
et al., 2022, 2023). Developing parsimonious models that maintain simplicity without sacrificing fidelity increases model
robustness, computational efficiency, and interpretability.
The effort needed to operationalize this activity—integrating PPEs into the model development cycle—should be seen in the
context of the current, suboptimal approach to model development. A well-designed PPE can be exploited by an entire
development team to consistently tackle biases. Barriers to wider adoption have been successfully tackled at several modelling
centres with streamlined PPE workflows (Elsaesser et al., 2025; Yarger et al., 2024).  We argue that operationalization could
enable a step-change in model evaluation and development that takes us beyond the current incremental advancements between
MIPs.

*We recommend developing a deeper understanding of how PPEs in combination with observations can be used to detect*
*structural deficiencies in models. This should include new methods, e.g., using machine learning and structurally diverse*
*models, to relate model-observation biases to deficiencies in the way that processes are represented in models.*



*We also recommend building PPEs into the model development cycle rather than being a separate activity after the tuned model version has been released. Such integration would enable model developers to understand the strengths and limitations of their parameterisation schemes and where model development is most needed and would be most effective.*


**2. Include PPEs in MIPs to link known uncertainties through to adaptation and mitigation decision making, to understand inter-model differences, and to lay the foundation for rigorous multi-model calibration.**

Multi-Model Ensembles (MMEs) are the main way that we communicate plausible uncertainties to wider science and impact communities. Adaptation, for example, is informed through a subset of CMIP simulations that are considered in IPCC's

Impacts, Adaptation and Vulnerability assessments (WG2). However, we lack a mechanism to capture the wider uncertainties associated with PPEs in current MMEs (Fig. 4a), even when modelling centres are aware of them. For example, Golaz et al. (2013) showed that there were multiple parameter configurations of their GFDL climate model consistent with observed present-day climate, but which simulated very different historical temperature evolutions. PPE ensembles subsequent to IPCC's 4th Assessment MIP (CMIP3) (Shiogama et al., 2014; Watanabe et al., 2012) showed that the CMIP3 MIROC3

configuration was right on the lower end of the 4-10 K PPE range of climate sensitivities consistent with MIROC3's formulation. Absence of this PPE context led to the impression of a more confident CMIP3 uncertainty range than supported by the known model uncertainties. There is likely to be similar tension for modelling centres for the upcoming CMIP7. For example, PPEs show how the development of the UK climate model has led to improvements in radiative fluxes compared to observations (Rostron et al., 2025), but with typically larger climate feedbacks than the previous model configuration (that

was deemed in CMIP6 to already be warmer than the IPCC's likely range). In the absence of a mechanism to capture PPE configurations in MMEs, insights from potential alternative configurations will not be captured in CMIP7 and will therefore not be visible to the wider science and impact communities.

Going beyond capturing alternative parametric configurations of models, a shift to systematic inclusion of PPEs within MMEs opens up potential for fuller, more rigorous multi-model calibration. A multi-model PPE (MMPPE) combined with emulators

to generate dense samples of model data (Fig. 2) would offer new opportunities to exploit statistical history matching (Craig et al., 1997; Lee et al., 2016; Williamson et al., 2013) or other calibration methods to observationally constrain climate metrics – Fig. 4b. In contrast, the unquantified mix of model structural, parametric and initial-condition uncertainty in MMEs makes it difficult to constrain the spread by down-weighting single models (Knutti et al., 2010) – Fig. 4c. History matching has been used successfully to constrain individual PPEs and would be readily extendable to multi-model PPEs regardless of whether

they followed a standardized protocol. Submission of a few PPE members to MIPs that span a range of model behaviours or climate metrics would be an efficient initial approach (Elsaesser et al., 2025; Hourdin et al., 2023; Peatier et al., 2022).





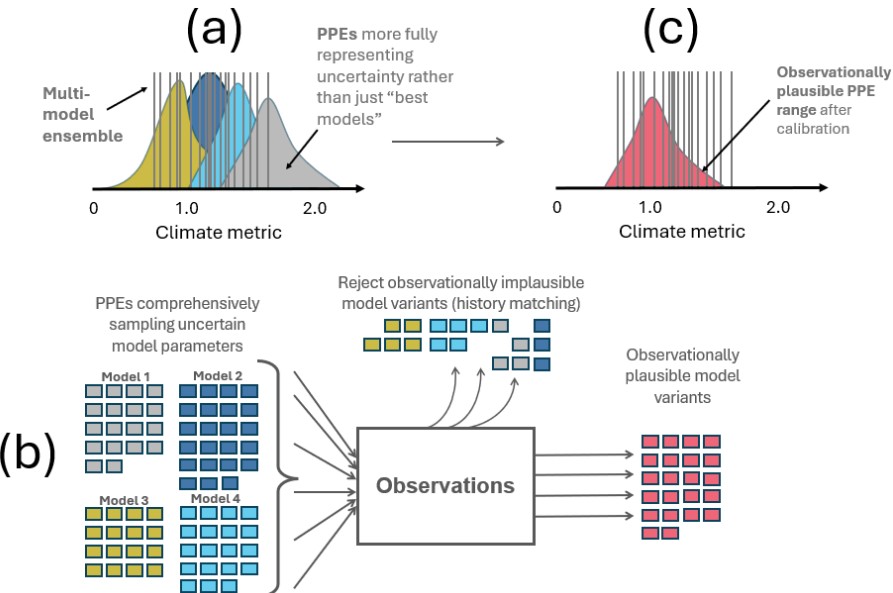

**Figure 4. Schematic illustrating two aspects of PPEs. (a) PPEs more fully represent uncertainties than a multi-model**
**ensemble (MME) of well-configured best estimates from several modelling centres. The MME (grey vertical lines)**
**represents an unquantified mixture of structural differences, parametric uncertainties and internal variability, making**
**it difficult to understand and ultimately reduce the spread in predictions or projections. PPEs (shown by pdfs) provide**
**a way of cleanly separating structural and parametric uncertainty. (b) Statistically rigorous model calibration can be**
**used to constrain the PPE range –shown in (c) – by defining a set of observationally plausible model variants, which is**
**not possible with an MME alone.**

*We recommend submission of selected PPE simulations to multi-model intercomparison projects to (i) remove the emphasis*
*on over-tuned "best" models so that plausible uncertainties can be more fully communicated to wider science and impacts*
*communities, (ii) more-comprehensively define model reliability, (iii) gain a deeper insight into the causes of multi-model*
*spread, and (iv) enable statistically rigorous calibration of MMEs against observations.*

*We also recommend efforts to separate internal variability and parametric uncertainty so that PPEs of models exhibiting*
*dynamic variability can be more easily compared. This could be achieved by running an initial condition ensemble in*
*combination with a PPE.*


**3. Adopt a common set of constraining observations and associated uncertainties.** While observations are extensively
used in MIPs (Waliser et al., 2020) to document and compare model skill, with MMPPEs they could also be used for model
calibration (Fig. 4). This would require systematic quantification of observational uncertainties and representation errors

(Johnson et al., 2020; Schutgens et al., 2017, 2016). The challenge is particularly large because inspection of individual
uncertainties is infeasible when comparing millions of model variants (estimates from emulators) against thousands of
observations in an automated statistical manner (Johnson et al., 2020). Reliable information on observation uncertainty is also
required in the detection of potential structural deficiencies (point 1) to avoid misattributing the causes of model-observations
biases.

*We recommend defining sets of observations for model constraint, including consideration of observation uncertainties and
representation errors in a form that would enable automated calibration of very large sets of model output.*

*We also recommend closer collaboration of the modelling and observational communities to understand the requirements of
model uncertainty quantification. This could include (i) efforts towards systematic measurement strategies in field campaigns*
*that consider the representativeness of the measurements* (Kahn et al., 2023) *and how they would be ingested into automated
model calibration; (ii) consideration of structural uncertainty in retrieval products through, for example, retrieval bundles*
(Chiu et al., 2024; Elsaesser et al., 2025).

**4. Account for and address model equifinality as a major cause of model unreliability.** Equifinality is where different
combinations of model parameters or structures produce simulations that cannot be distinguished within the uncertainty of
observations (Beven, 2006), but may diverge and cause large uncertainty in predictions or retrodictions (e.g., projections or
historical climate simulations – see Fig. 5). In one example related to aerosols (Lee et al., 2016), equifinality was so strong
that observations with arbitrarily small uncertainty did not constrain radiative forcing. Equifinality has profound implications
for our interpretation of model performance (how well models match observations), which does not guarantee reliable
historical or future simulations (Golaz et al., 2013). Without a PPE, it is impossible to assess the scale of this problem.
Research is needed to identify the model processes responsible for compensating effects so that strategies can be developed to
reduce their effect on model constraint. There are ways forward given that equifinality often occurs because observations of
state variables (such as aerosol concentration) do not constrain the compensating processes, which we highlight in point 5.

*We recommend research to understand the causes of error compensation (equifinality) and ways to minimize the effects.*

*We also recommend submission of equifinal model variants developed using PPE methods to future MIPs to better characterize
model reliability and to understand which projection metrics are most affected by equifinality.*



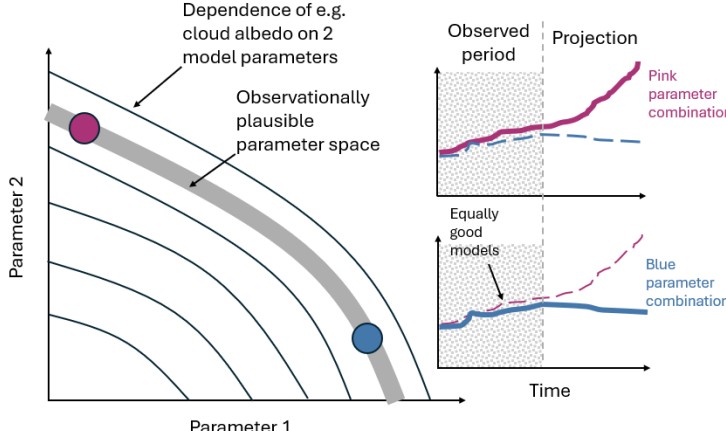


**Figure 5. Schematic of how equifinality affects model reliability. During the period with observations (stippled), the pink and blue parameter combinations have similar model performance and appear to be reliable, but they can diverge and become unreliable in projections. Poor model projection reliability can therefore be masked when simulating only one of the parameter combinations, but will become visible with a PPE.**


**5. Use PPEs to identify new observations and new ways of using observations to constrain models.** PPEs are a valuable tool to not only leverage existing observations better, but also to plan future observations targeted at model constraint. This objective was highlighted by community consensus in the US-CLIVAR sponsored workshop Micro2Macro: Origins of Climate Change Uncertainty (McCoy et al., 2025). There are opportunities, for example, to understand where observations are effective
or ineffective at reducing model uncertainty, the processes that cause the remaining uncertainty, and which observations would help to reduce it (Regayre et al., 2025). There are also opportunities to use observations in more creative ways, such as observational constraint of model processes rather than just simulated states, which would help to reduce equifinality. Ultimately, we observe states, not processes, and we must stitch these states together with models to infer causation (Feingold et al., 2025; Hume, 1751). Outside of a few cases (Christensen et al., 2022; Malavelle et al., 2017; McCoy et al., 2018), it is
very hard to infer causality from observations of state because of equifinality -- many combinations of processes can yield the same state (Gryspeerdt et al., 2019; Stevens and Feingold, 2009; Wood et al., 2012). PPEs provide an opportunity to infer causation from observations (Mikkelsen et al., 2025b).

*We recommend using PPEs to develop ways of identifying new observations or new combinations of existing observations with*
*the greatest potential to constrain model uncertainty.*

*We also recommend exploiting PPEs to infer causation and process-level understanding from observations of atmospheric state.*



**6. Exploit PPEs of hierarchies of models to understand system behaviour and develop parameterizations.**
Parameterization development typically involves a hierarchy of models and several steps, including understanding key process interactions across diverse environmental conditions, conducting sensitivity tests, and performing optimization and simplification (Gettelman, 2023). PPEs are ideal for tackling all these steps. PPEs of high-resolution process-based models have been performed in recent years and, as described above, they have helped to understand the joint effects of co-varying

environmental conditions (Park et al., 2020; Sansom et al., 2024, 2025; Wellmann et al., 2020). Early research on connecting several models is very promising (Couvreux et al., 2021; Hourdin et al., 2021) and there is considerable scope to test or develop regime-aware parameterizations in a more comprehensive way than is possible with single perturbation studies or intercomparisons of process-based models (Blossey et al., 2016; Zhang et al., 2013).

*We recommend applying PPEs across hierarchies of multi-scale models with varying levels of process sophistication to enhance our understanding of complex, nonlinear processes and to develop parameterisations for large-scale models.*

## 5. Conclusions

In summary, we argue that PPEs:

1. Improve our understanding and quantification of model uncertainty in the widest sense by cleanly separating structural
and parametric causes of uncertainty.

2. Provide deep insight into the causes of multi-model spread.

3. More fully characterize the plausible spread in climate projections, which is vital to better communicate current knowledge to downstream science, impacts and decisions.

4. Generate rich datasets that can be used for statistically rigorous observational calibration of model outputs to better
characterize and reduce multi-model spread.

5. Guide the judicious selection of model structural developments or simplifications.

6. Guide observational strategies and ways of using existing observations to achieve the greatest possible constraint of models.

7. Provide a way to disentangle the interacting environmental drivers of system behaviour and to infer causation from
observations of atmospheric state.

8. Contribute to the development of parameterizations by linking processes and sensitivities across a hierarchy of models.

We started this opinion piece by pointing out that there are several essentially competing efforts in climate modelling – complexity, resolution and initial condition ensembles. To this list we add perturbed parameter ensembles, which we argue are making a substantial contribution to assessing and improving model reliability, which is vital given the high societal cost of

model uncertainty (Hope, 2015). Increased model complexity and resolution will undoubtedly provide additional detailed and

actionable information, for example about some types of extreme or local events. However, the development of climate models over several decades has shown that, while our understanding of fundamental processes (knowledge uncertainty) has improved, we have not translated this improved knowledge into commensurate improvements in our understanding of model reliability. This lack of translation is what Box meant by "The scientist cannot obtain a 'correct' [model] by excessive elaboration" (Box, 395 1976), but we argue that PPEs provide the best opportunity to obtain models that are as 'correct' as possible. Critically, applying PPEs as outlined here would enable more robust estimates of model uncertainty, empowering stakeholders to realistically evaluate risk and make informed decisions based on information from the Earth system modelling community, rather than relying on a single high-fidelity model variant. Treading a new path, with increased emphasis on the use of PPEs in model evaluation and development will help to address the model uncertainty challenge.

**Acknowledgements**

The authors acknowledge funding from NERC (NE/X013901/1 and NE/Y001028/1); EU (821205); Research Council of Finland (359166); Department of Energy (DOE) (DE-SC0024161, DE-SC0021270, DE-SC0022323, and DE-SC0023151); the Transregional Collaborative Research Center SFB/TRR 165 "Waves to Weather", the Italia—Deutschland science-4-services network in weather and climate (4823IDEAP6) funded by the German Federal Ministry of Digital and Transport; 405 NASA Model Analysis and Prediction program (MAP) (#80NSSC21K1498), the NSF STC Learning the Earth with Artificial Intelligence and Physics (LEAP) (2019625 and AGS-2203001); NASA (MAP NNH16ZDA001); Swiss National Science Foundation Post Doc Mobility Grant (217899). LR, DS and KY were supported by the Met Office Hadley Centre Climate Programme funded by DSIT. The Pacific Northwest National Laboratory is operated for the U.S. Department of Energy by the Battelle Memorial Institute under contract DE-AC05-76RL01830.

**Author contributions**

All authors contributed to the conceptualization and writing.

**Competing interests**

KC, AO and YQ are members of the ACP editorial board.

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
