# Peer review of "Opinion: The importance and future development of perturbed parameter ensembles in climate and atmospheric science"

_EGUsphere, 2025_

## Referee Comment (RC1)

The authors are to be commended for condensing what was invariably a great many ideas into a coherent and well structured perspective. Their manuscript brings together a wealth of experience to help explain why they find perturbed parameter ensembles to be useful. I also liked the selection of Figures, which were well constructed and informative. The Authors' views, presented in this manner, will be of general interest to the community and serve as a touchstone for further discussions. I also found no issues with their Perspective's reading of the literature, in which sense I think the article is scientifically sound. I would thus simply offer the authors the benefit of my views, which they can respond to as they see fit, but which need not entail changes to their manuscript.

I'd begin with a few overarching comments:

The first is that the manuscript leaves it to the reader to infer the authors' view of modelling. My inference is that they think it is to provide an instrument to guide policy, i.e., the authoritative view, a crystal ball. But there are other views, and these have an implication for how one interprets the manuscript. It would be helpful if these views were more in the forefront when presenting the authors ideas.

My view of modelling is that it is a tool we create to advance our reasoning. Models aren't for prediction, rather predictions are what we do to test our reasoning and this reasoning is informed by the use of models. This leads to my favored interpretation of the manuscript, as an elaborated way of saying that people should play with their models to understand why they do what they do and use this to assess the reasoning that the models helped develop and which can't be more directly tested.

Another view, is that the point of modelling is to establish authority (reliability measures this). This sounds negative, and it is, but it is also necessary in practice, for instance to develop policy. Hence it is also a valid view, which the manuscript appears to adopt It then proceeds to give the impression that PPEs can be coordinated in a formal way, to better establish the authority (reliability) of a model, rather than its suitability as a tool for developing an argument.

This leads to my second overarching point. Assuming that we are talking about models as tools for prediction, then I don't see how the author's ideas could be implemented. The paper certainly presents many arguments as to why their ideas have merit But it assumes that they could be implemented to improve the practical use of models. I think this assumption is false.

My skepticism has a couple of origins. One, is that the ideas are not new, and such approaches have not worked in the past, at least not as applied to Earth system models. So why would they work now? To address this it could help to consider if past failures occurred for structural reasons, i.e., related to the fact that the models (at least the parameterizations) aren't really physical, but rather structurally different statistical fits, to a developing understanding of the data. Or were past failures for cultural reasons, i.e., we lack the discipline and organizational skill needed to implement the approach. And if one or the other, what do we need to do differently, and why are we capable of doing so.

Part of the difficulty is this idea of matching observations. In model development we generally have an idea of when a simulation is closer to some observable quantity as compared to another simulation, and this often guides both structural and parametric choices. However in Earth system modelling, it almost never happens that a change to a model is closer to all observations. Likewise it almost never happens that the same change across all models gives the same improvement. It would seem that both are required for the author's programme to make sense, and neither is. The better the example of how my thinking is wrong, the more optimistic I would be about the proposed research programme.

To address the above it would be helpful to work out one or two examples of how the author's views could be implemented. The detail and specificity of the examples will be important, as until now all one has are vague references to studies that point in a given direction. Some questions that should be addressed by the examples would be: How do the results depend on what parameters are chosen? Models have thousands of parameters. Is the idea to expose them all and vary them all? Is that possible? Is it worth it? How to deal with the fact that similarly named parameters have different meanings in different models, and that many parameters are hidden? How to determine plausible parameter ranges of unphysical parameters? How many models are needed for this programme to work? How to deal with the fact that structural uncertainty is

grossly undersampled (we all more or less use the same model, e.g., Shaw and Stevens, *Nature*, 2025). And most importantly, what would the result look like at the end?

Some more specific comments:

- In section 3 I would have welcomed a more specific discussion of what was learned, rather than what was done.  Learning generalizes.

- Despite the definition of terms, which I very much liked, the authors used the word uncertainty quite loosely.   Also, if a parameterization is based on a false assumption, how can the parameters it uses have a correct value, and if they can't then what does parameter uncertainty really mean.  This all seems predicated on the idea that the model is structurally correct and we know it is not.  Hence I think it is not correct to think of structural and parametric uncertainty, but rather one should speak of structural and parametric *sensitivities*.

- Mauritsen et al., (*J. Adv. Model. Earth Syst.* 2012) was the first study to openly discuss the strategy of tuning a CMIP model, and it encountered quite a bit of resistance from our own community who felt that this was opening our field to disingenuous criticism.   The strategy we adopted at that time was very much cognizant of the idea of equifinality.  Hourdin et al., (2012) is similar and came out at the same time.   The later Hourdin et al., article on the "Art and Science" was an outgrowth of these more foundational contributions (which incidentally was initiated by S. Bony, who recognized the importance of these ideas and, as these things go, wasn't every properly acknowledged.)

- For Fig 4.  How does one reject implausible estimates?  Somehow this happens magically in the box labeled observations, but it is the crux of the matter.  If this were possible then it should also be possible with the MME and that raises two questions:  why hasn't it been done in the past (i.e., in IPCC assessments of the CMIP ensemble); and whether the best estimate would end up being different? In other words maybe MMEs adequately sample parameter and structural sensitivities.

Bjorn Stevens

2025-10-04, Berlin

---

## Author Comment (AC1)

We appreciate the useful comments. Reviewer comments are in black and our responses in blue, with **bold** indicating where we have made a change to the paper.

The opinion paper by Carslaw et al. presents a well layed-out argumentation for leaving space for a PPE component in future model development. It presents a very interesting overview of PPE work in the field and I appreciate the effort of the authors to land on several recommendations for how to use PPEs. This paper serves its purpose to provide an "opinion" in the field of climate model development and beyond.

My suggestions for improving the paper are minor:

To the end of the abstract the authors mention, that PPEs should be prioritized when allocating computing resources. I am not sure what prioritize here means. Giving 1st priority to PPEs is to my opinion a too strong wording, since other methods and workflows using climate models may claim priority for good reasons. Preparing a bug-free, multi-purpose ESM model code may need considerable computing resources, often not appreciated by funding agencies neither. And, while having great potential, PPEs as such do not remove structural error or provide scenario simulations for different futures. Not all problems require a PPE. See also another "opinion" paper by Jones et. al, https://doi.org/10.5194/esd-15-1319-2024, on the use of ESMs and improved cooperation to develop them. A word of caution when mentioning priorities would be to the advantage of the paper.

We appreciate this comment. We decided to just delete this statement about priority.

Missing is possibly also a discussion of the challenges of PPEs. Why haven't PPEs been used more often? There are obstacles for that. The PPE implementation in models, efficient launch scripts, the large amount of data, the demanding handling of a lot of data, the waste of resources on implausible model variants, all these are challenges. But I support very much that model development teams should consider the use of PPEs in their model development workflow, as opposed to only one-at-a-time-testing of parameter choices for tuning and model improvement.

**We have added some text on this in the paragraph on point 1 where we recommend operationalisation of PPEs. This is where we said that barriers to wider adoption have been tackled... We now say:**

"*There have been several barriers to wider adoption of the PPE approach, such as the challenge of selecting appropriate parameters and perturbation ranges (which often requires input from multiple developers), designing the simulations, implementing the perturbations in models, defining efficient workflows and simulation submission scripts, and the large data volumes that are produced. However, we estimate that PPEs have now been developed across a wide range of model types by over twenty research groups in small research teams and large modelling centres, which has reduced some of the knowledge barriers. Barriers to wider operationalisation have been successfully tackled at several modelling centres with streamlined PPE workflows (Elsaesser et al., 2025; Yarger et al., 2024).*"

Line 225 invites for asking me to add another word of caution: "PPEs provide the only means to disentangle structural and parametric causes of model–observation biases". Single process investigation, varying one at a time parameter variation, has been used efficiently in the past to include more correct and important processes in climate models. Evaluation with multiple

observations has been shown to be useful to find structural uncertainty of models, without a PPE.

Yes, this is correct. We could add that it's the only way to *rigorously* disentangle the causes. Perturbing one parameter at a time may or may not identify the best new process to include. In some cases, the link between a model-observation bias and a missing process might be straightforward, but there are any cases where this is not the case. For example, in the aerosol world, too-low particle number may be caused by missing nucleation mechanisms or by incorrect sinks. But we agree that a PPE is not the only way.

**We have rephrased this sentence to say "PPEs provide a very effective way to disentangle…"**

Another "missing":

The importance of parameterisation documentation. When exploring model diversity and comparing model sensitivities across models, understanding the code differences and details of parameterisation choices has been a long standing challenge in all MIPs. It becomes even more important when doing multi-model PPEs. How to do such documentation efficiently is still a challenge. It might be, though, a positive side effect of organising multiple model PPEs, that such documentation becomes more clear, apparent and accessible for understanding model differences. The authors state in the conclusion: "We started this opinion piece by pointing out that there are several essentially competing efforts in climate modelling – complexity, resolution and initial condition ensembles. To this list we add perturbed parameter ensembles." I believe transparent model documentation should be added to this list, in particular when thinking also of human resources needed to do the modelling.

This is a good point, although we don't think the best place for this is in the coda of the paper.

**We have added a sentence in recommendation 2 related to MMPPEs: "***Multi-model PPEs (MMPPEs) will require an improved level of parameterization documentation, although efforts to design MMPPEs may naturally bring this about.***"**

---

## Author Comment (AC2)

We appreciate the reviewer's perspectives. Reviewer comments are in black and our responses in blue, with **bold** indicating where we have made a change to the paper.

The authors are to be commended for condensing what was invariably a great many ideas into a coherent and well structured perspective. Their manuscript brings together a wealth of experience to help explain why they find perturbed parameter ensembles to be useful. I also liked the selection of Figures, which were well constructed and informative. The Authors' views, presented in this manner, will be of general interest to the community and serve as a touchstone for further discussions. I also found no issues with their Perspective's reading of the literature, in which sense I think the article is scientifically sound. I would thus simply offer the authors the benefit of my views, which they can respond to as they see fit, but which need not entail changes to their manuscript.

I'd begin with a few overarching comments: The first is that the manuscript leaves it to the reader to infer the authors' view of modelling. My inference is that they think it is to provide an instrument to guide policy, i.e., the authoritative view, a crystal ball. But there are other views, and these have an implication for how one interprets the manuscript. It would be helpful if these views were more in the forefront when presenting the authors ideas.

My view of modelling is that it is a tool we create to advance our reasoning. Models aren't for prediction, rather predictions are what we do to test our reasoning and this reasoning is informed by the use of models. This leads to my favored interpretation of the manuscript, as an elaborated way of saying that people should play with their models to understand why they do what they do and use this to assess the reasoning that the models helped develop and which can't be more directly tested.

Another view, is that the point of modelling is to establish authority (reliability measures this). This sounds negative, and it is, but it is also necessary in practice, for instance to develop policy. Hence it is also a valid view, which the manuscript appears to adopt. It then proceeds to give the impression that PPEs can be coordinated in a formal way, to better establish the authority (reliability) of a model, rather than its suitability as a tool for developing an argument.

We didn't intend to give the impression that PPEs are primarily about authority and useful only for policy purposes. This seems more the purpose of Earth Visualisation Engines (Stevens et al., 2024). Our conclusion begins with a summary of the ways PPEs are being used to deepen our understanding of model skill and behaviour. One of the main points of the article is that PPEs are a powerful and increasingly efficient tool to probe and understand model behaviour (or to use Bjorn's words, "playing with a model"). For example "*We recommend exploiting PPEs to infer causation and process-level understanding from observations of atmospheric state.*" Also "*We recommend applying PPEs across hierarchies of multi-scale models with varying levels of process sophistication to enhance our understanding of complex, nonlinear processes and to develop parameterisations for large-scale models*".

**We have amplified the roles of PPEs in the final paragraph of the introduction. We said:**

"In this article, we highlight how PPE science has expanded since the 2015 Workshop on Uncertainty Quantification in Climate Modeling and Projection (Qian et al., 2016) to address the overall challenge of model uncertainty and to provide an increasingly deep, process-level understanding of model behaviour. In particular, we argue that PPEs can help to more fully characterize current model uncertainty and provide a path toward reducing it…"

We now say

*"In this article, we highlight how PPE science has expanded and diversified since the 2015 Workshop on Uncertainty Quantification in Climate Modeling and Projection (Qian et al., 2016). PPEs are now being extensively used to more fully characterize model uncertainty and to provide a path toward reducing it. However, PPEs are no longer just applied to questions of model uncertainty and reliability; they are also now being used to develop a deeper, process-level understanding of model behaviour..."*

**We have also added a reference to the Earth Visualization Engine paper (Stevens et al., 2024), which I think nicely draws a distinction between what the reviewer believes is important and what we are discussing in our Opinion.**

This leads to my second overarching point. Assuming that we are talking about models as tools for prediction, then I don't see how the author's ideas could be implemented. The paper certainly presents many arguments as to why their ideas have merit. But it assumes that they could be implemented to improve the practical use of models. I think this assumption is false.

My skepticism has a couple of origins. One, is that the ideas are not new, and such approaches have not worked in the past, at least not as applied to Earth system models. So why would they work now? To address this it could help to consider if past failures occurred for structural reasons, i.e., related to the fact that the models (at least the parameterizations) aren't really physical, but rather structurally different statistical fits, to a developing understanding of the data. Or were past failures for cultural reasons, i.e., we lack the discipline and organizational skill needed to implement the approach. And if one or the other, what do we need to do differently, and why are we capable of doing so.

It is premature to say PPEs "haven't worked" because they have only recently been used extensively enough to evaluate the breadth of their usefulness. We provide ample examples. They have worked in a traditional sense by, for example, informing the spread of climate impact projections (e.g., dataset at https://ukclimateprojections-ui.metoffice.gov.uk/products/LS2_Subset_02). There now are as many groups seriously exploring PPEs (about 28 groups listed in this Opinion) as there are exploring the benefits of very high resolution (see the many references in section 3), so we can expect some acceleration of the benefits.

There are a few reasons why PPEs have not been routinely used to improve model predictions:

- First, modelling groups have historically used their finite resources (people and computing) to contribute to MIPs and to enhance model complexity and resolution. This was obviously a sensible strategy because models lacked some key processes and had relatively coarse resolution. Such an approach also exploited skill sets of the kinds of scientist who became involved in climate science over the last two or three decades (physical scientists, but rarely statisticians). As our article shows, many groups are now increasingly looking to PPEs to tackle a range of complex problems.
- Second, we agree that there may be cultural reasons that helped entrench the paradigm that more complex models are intrinsically skilful. This approach has not reduced model uncertainty and has increased model diversity. However, the breadth of climate research utilizing PPEs we outline in the article suggests this paradigm is increasingly seen as outdated.

- Third, the PPE approach is technically and statistically challenging, requiring specialised workflows that may historically have been a barrier to uptake. Now that these workflows are fairly standard, typically openly accessible and optimised for efficiency, they can be applied much more easily by other groups.

Part of the difficulty is this idea of matching observations. In model development we generally have an idea of when a simulation is closer to some observable quantity as compared to another simulation, and this often guides both structural and parametric choices. However in Earth system modelling, it almost never happens that a change to a model is closer to all observations. Likewise it almost never happens that the same change across all models gives the same improvement. It would seem that both are required for the author's programme to make sense, and neither is. The better the example of how my thinking is wrong, the more optimistic I would be about the proposed research programme.

> Commented [LR1]: Need to comment on this. MM-PPEs are a way forward. Otherwise, models become more skillful at composite observations like AOD whilst diverging in specific properties such as aerosol species concentration (Gliss et al., 2021)

The utility of PPEs does not rely on either of these assumptions. We absolutely agree that changes to a model almost never bring it into closer agreement with *all* observations, nor that the same change produces the same improvement across all models. In fact, these difficulties are motivations for using PPEs rather than traditional approaches to model development. You can explore the reasons for the inconsistencies.

The point we made is that the failure to agree with all observations probably indicates that the cause is a structural deficiency in the model (having exhausted the effects of joint perturbations of parameters). The study by Regayre et al. (2023) showed this, followed up by Prevost et al. (2025). From an individual model perspective, of course we want our models to agree with lots of observations, but it's no more of a requirement of PPEs than traditional modelling approaches which laboriously explore one process at a time to achieve this goal. From a multi-model perspective, we want to understand why models respond distinctly to similar perturbations which is why we recommend creating and analysing multi-model PPEs which would allow us to systematically identify areas of agreement and divergence, and the processes that affect them.

To address the above it would be helpful to work out one or two examples of how the author's views could be implemented. The detail and specificity of the examples will be important, as until now all one has are vague references to studies that point in a given direction. Some questions that should be addressed by the examples would be: How do the results depend on what parameters are chosen? Models have thousands of parameters. Is the idea to expose them all and vary them all? Is that possible? Is it worth it? How to deal with the fact that similarly named parameters have different meanings in different models, and that many parameters are hidden? How to determine plausible parameter ranges of unphysical parameters? How many models are needed for this programme to work? How to deal with the fact that structural uncertainty is grossly undersampled (we all more or less use the same model, e.g., Shaw and Stevens, Nature, 2025). And most importantly, what would the result look like at the end?

These points feel like a common riposte: "you've developed a more-comprehensive approach to modelling, but it won't deliver the desired results unless you push it to the limit of non-achievability". We could equally well ask the same of developments in complexity or resolution - surely they will only pay out once you've reached 1 metre resolution or added 1000 new processes.

Let's answer the specific questions:

How do the results depend on what parameters are chosen? This is an important and open question. At present there are too few PPEs to answer this question, which is why wider uptake and coordination is needed. Existing PPE studies show that a relatively small number of parameters control the spread/uncertainty of any chosen model output (next point), so the key to success is including the most important parameters, which are more readily identified. Comparing PPEs across models with the similar perturbations will help to understand why models behave similarly or dissimilarly.

Models have thousands of parameters. Is the idea to expose them all and vary them all? Is that possible? Is it worth it? All of the modelling challenges that we referred to in the Opinion (uncertainty, complexity, resolution, variability) share a common challenge – finding the appropriate level of completeness to answer the questions posed. Yes, there are thousands of parameters in a climate model, but scoping exercises (usually one-at-a-time perturbations) show that usually less than 10 affect large-scale behaviour and only around 50 or 60 really matter overall (usually regionally only). Often there are many parameters in a parameterization, but often it is the process rate that is treated as "the parameter", which therefore reduces the number in a PPE. As with all modelling strategies, the goal is not unobtainable completeness, but sufficiency for the question at hand.

How to deal with the fact that similarly named parameters have different meanings in different models, and that many parameters are hidden? There's no expectation that all models would perturb the same parameters, just like it is not necessary that all models include the same processes (in fact the latter makes the former impossible). Diversity of PPE design is expected and absolutely fine. Progress will be made, not by seeking parameter-level uniformity, but by exploring how uncertainty in key processes manifests in each model and how those manifestations compare across models.

How to determine plausible parameter ranges of unphysical parameters? Unphysical parameters are a problem for climate modelling in general, and not just PPEs. Often, as in the case of the Zhu et al. paper you mention in your addendum review, inclusion of such parameters probably reflects the need to account for model structural deficiencies. The benefit of a PPE is that creating one forces model developers to deeply evaluate their assumptions, which can help to expose such parameters. Such parameters can be calibrated in the same way as physical parameters, using observations. Maybe the resulting plausible range of an unphysical parameter will at least raise some questions for the developers.

How many models are needed for this programme to work? As with multi-model ensembles, this is a sliding scale. "To work" can mean many things. There is limited value in defining a minimum number, since we can gain huge insight from just one or two PPEs. The aim is to build insight progressively as the number of PPEs increases.

How to deal with the fact that structural uncertainty is grossly undersampled? We agree that quantifying structural uncertainty is a challenge, but this applies equally to MMEs and PPEs. Is structural uncertainty grossly undersampled? We have no concrete evidence of that. If key observations were persistently outside the PPE range of multiple models, we would have evidence of structural deficiencies that would help target future development. PPEs provide a path to identifying the deficiencies in a rigorous way (Regayre et al., 2023, Prevost et al., 2025). In contrast, increasing model complexity because we *think* the model has structural deficiency

is the path to overly complex, over-determined models that suffer from equifinality and are harder, not easier, to verify against observations.

What would the result look like? The outcome will not be a single "best" model, but a clearer understanding of which processes, parameters and structural choices control model skill and behaviour. We expect greater insight into which uncertainties matter most for different applications and greater clarity around when structural changes are genuinely required. The knowledge gained through PPE analyses can guide model development, observational priorities, and interpretation of projection uncertainties in a way that is difficult to achieve with traditional approaches.

Some more specific comments:

In section 3 I would have welcomed a more specific discussion of what was learned, rather than what was done.  Learning generalizes.

We have considered this suggestion, but we don't want to turn this Opinion into a full review. Nevertheless, we could point to a few places where we already highlight what was achieved by using PPEs, including: Understanding and constraining uncertainty in climate sensitivity, constraining climate projections, exposing model structural deficiencies, exposing how combinations of cloud-controlling factors affect cloud evolution, understanding how environmental conditions and uncertainties in cloud microphysics affect the evolution of weather-scale phenomena, and guiding model development for improved tracking of wind and solar energy. There are many more examples.

Despite the definition of terms, which I very much liked, the authors used the word uncertainty quite loosely.

It seems useful to define uncertainty directly in the table as:

**"Model uncertainty is the range of possible outcomes that results from incomplete or imperfect representation of the climate system in the model, including processes, inputs and assumptions."**

If a parameterization is based on a false assumption, how can the parameters it uses have a correct value, and if they can't then what does parameter uncertainty really mean.

This all seems predicated on the idea that the model is structurally correct and we know it is not.  Hence I think it is not correct to think of structural and parametric uncertainty, but rather one should speak of structural and parametric sensitivities.

We agree that models are not structurally correct in any absolute sense, and we do not assume they are. However, we do not think this precludes meaningful discussion of parametric uncertainty, nor does it require that we restrict ourselves to discussing sensitivity.

Uncertainty is not equivalent to sensitivity. Sensitivity describes how model outputs respond to parameter changes, whereas uncertainty combines that sensitivity with the likely parameter range. Even when a parameter is based on an imperfect or simplified representation of reality, its parameters typically have plausible limits that are informed by theory, expert knowledge, and laboratory measurements. These parameter bounds can be further refined using observational constraints that explicitly account for measurement uncertainty. In this sense, parameter

uncertainty (and the associated model output uncertainty) remain meaningful even when the underlying assumptions are known to be approximate.

Traditional modelling approaches assume sufficient structural adequacy for tuning to be meaningful. Otherwise, model tuning would be considered pointless and potentially misleading. In contrast, as we point out in the Opinion, PPE analyses are not predicated on the assumption of structurally correct models, which a lot of people argue is the case. They are now part of a wider research activity that can iteratively converge on structurally sound models with quantifiable uncertainty. PPEs provide a way to avoid overconfidence in tuned models and a path towards structurally better models.

Mauritsen et al., (J. Adv. Model. Earth Syst. 2012) was the first study to openly discuss the strategy of tuning a CMIP model, and it encountered quite a bit of resistance from our own community who felt that this was opening our field to disingenuous criticism. The strategy we adopted at that time was very much cognizant of the idea of equifinality. Hourdin et al., (2012) is similar and came out at the same time. The later Hourdin et al., article on the "Art and Science" was an outgrowth of these more foundational contributions (which incidentally was initiated by S. Bony, who recognized the importance of these ideas and, as these things go, wasn't every properly acknowledged.)

Indeed, **we now add this reference**. There it was referred to as error compensation I think. It wasn't fully clear in that paper whether it was error compensation or error "acceptance". For example, it discussed tuning TOA energy balance while accepting errors in the cloud processes. This is different to what we mean by equifinality, which is multiple model set-ups that achieve equally good model performance.

• For Fig 4. How does one reject implausible estimates? Somehow this happens magically in the box labeled observations, but it is the crux of the matter. If this were possible then it should also be possible with the MME and that raises two questions: why hasn't it been done in the past (i.e., in IPCC assessments of the CMIP ensemble); and whether the best estimate would end up being different? In other words maybe MMEs adequately sample parameter and structural sensitivities.

There is no single way of identifying implausible models. It will always rely on the information available, such as uncertainty in observations, emulators, etc. One way is to use the implausibility metric, which weights the difference between the model and measurements by the uncertainties associated with the comparison (Craig et al., 1996; Williamson et al., 2013; Johnson et al., 2020).

**We now refer to the implausibility metric in the caption of Fig 4 and have added this citation.**

However, the key point here is that this approach cannot be used reliably for an MME. If there are, say, 50 important parameters and dozens of alternative model structures, then a set of, say, 30 MME members is extremely sparse in that >50-dimensional space. This is statistically impossible (it's like screening just 30 people for diseases when there are 50 compounding potential causes). But it is feasible with PPEs when combined with emulators. They can generate millions of surrogate model values, or whatever number produces sufficiently dense sampling of the 50-dimensional space. This is the problem with MMEs: we just don't have enough members to do statistics when there are such a large number of factors that cause differences between them.

**We have altered the wording to make it clearer that MMEs cannot be used in this way. The previous text was**

**"In contrast, the unquantified mix of model structural, parametric and initial-condition uncertainties in an MME makes it difficult to constrain the spread by down-weighting single models [1] – Fig. 4c."**

**We now say "*In contrast, the unquantified mix of dozens of model structural, parametric and initial-condition uncertainties represented by an extremely small number of MME members makes it statistically inappropriate to constrain the spread by down-weighting single models* [1] – *Fig. 4c."***

**References**

Craig, P. S., Goldstein, M., Seheult, A. H., and Smith, J. A.: Bayes linear strategies for history matching of hydrocarbon reservoirs, in: Bayesian Statistics 5, edited by: J. M. Bernado, J. O. Berger, A. P. Dawid, and A. F. M. Smith, 69–95, Clarendon Press, Oxford, UK, 1996.

Johnson, J. S., Regayre, L. A., Yoshioka, M., Pringle, K. J., Turnock, S. T., Browse, J., Sexton, D. M. H., Rostron, J. W., Schutgens, N. A. J., Partridge, D. G., Liu, D., Allan, J. D., Coe, H., Ding, A., Cohen, D. D., Atanacio, A., Vakkari, V., Asmi, E., and Carslaw, K. S.: Robust observational constraint of uncertain aerosol processes and emissions in a climate model and the effect on aerosol radiative forcing, Atmos. Chem. Phys., 20, 9491–9524, https://doi.org/10.5194/acp-20-9491-2020, 2020.

Prévost, L. M. C., Regayre, L. A., Johnson, J. S., McNeall, D., Milton, S., and Carslaw, K. S.: Detection of structural deficiencies in a global aerosol model to explain limits in parametric uncertainty reduction, EGUsphere [preprint], https://doi.org/10.5194/egusphere-2025-4795, 2025.

Regayre, L. A., Deaconu, L., Grosvenor, D. P., Sexton, D. M. H., Symonds, C., Langton, T., Watson-Paris, D., Mulcahy, J. P., Pringle, K. J., Richardson, M., Johnson, J. S., Rostron, J. W., Gordon, H., Lister, G., Stier, P., and Carslaw, K. S.: Identifying climate model structural inconsistencies allows for tight constraint of aerosol radiative forcing, Atmos. Chem. Phys., 23, 8749–8768, https://doi.org/10.5194/acp-23-8749-2023, 2023.

Stevens, B., Adami, S., Ali, T., Anzt, H., Aslan, Z., Attinger, S., Bäck, J., Baehr, J., Bauer, P., Bernier, N., Bishop, B., Bockelmann, H., Bony, S., Brasseur, G., Bresch, D. N., Breyer, S., Brunet, G., Buttigieg, P. L., Cao, J., Castet, C., Cheng, Y., Dey Choudhury, A., Coen, D., Crewell, S., Dabholkar, A., Dai, Q., Doblas-Reyes, F., Durran, D., El Gaidi, A., Ewen, C., Exarchou, E., Eyring, V., Falkinhoff, F., Farrell, D., Forster, P. M., Frassoni, A., Frauen, C., Fuhrer, O., Gani, S., Gerber, E., Goldfarb, D., Grieger, J., Gruber, N., Hazeleger, W., Herken, R., Hewitt, C., Hoefler, T., Hsu, H.-H., Jacob, D., Jahn, A., Jakob, C., Jung, T., Kadow, C., Kang, I.-S., Kang, S., Kashinath, K., Kleinen-von Königslöw, K., Klocke, D., Kloenne, U., Klöwer, M., Kodama, C., Kollet, S., Kölling, T., Kontkanen, J., Kopp, S., Koran, M., Kulmala, M., Lappalainen, H., Latifi, F., Lawrence, B., Lee, J. Y., Lejeun, Q., Lessig, C., Li, C., Lippert, T., Luterbacher, J., Manninen, P., Marotzke, J., Matsouoka, S., Merchant, C., Messmer, P., Michel, G., Michielsen, K., Miyakawa, T., Müller, J., Munir, R., Narayanasetti, S., Ndiaye, O., Nobre, C., Oberg, A., Oki, R., Özkan-Haller, T., Palmer, T., Posey, S., Prein, A., Primus, O., Pritchard, M., Pullen, J., Putrasahan, D., Quaas, J., Raghavan, K., Ramaswamy, V., Rapp, M., Rauser, F., Reichstein, M., Revi, A., Saluja, S., Satoh, M.,

Schemann, V., Schemm, S., Schnadt Poberaj, C., Schulthess, T., Senior, C., Shukla, J., Singh, M., Slingo, J., Sobel, A., Solman, S., Spitzer, J., Stier, P., Stocker, T., Strock, S., Su, H., Taalas, P., Taylor, J., Tegtmeier, S., Teutsch, G., Tompkins, A., Ulbrich, U., Vidale, P.-L., Wu, C.-M., Xu, H., Zaki, N., Zanna, L., Zhou, T., and Ziemen, F.: Earth Virtualization Engines (EVE), Earth Syst. Sci. Data, 16, 2113–2122, https://doi.org/10.5194/essd-16-2113-2024, 2024.

Williamson, D., Goldstein, M., Allison, L., Blaker, A., Challenor, P., Jackson, L., and Yamazaki, K.: History matching for exploring and reducing climate model parameter space using observations and a large perturbed physics ensemble, Clim. Dynam., 41, 1703–1729, https://doi.org/10.1007/s00382-013-1896-4, 2013.